# MODELING WINNER-TAKE-ALL COMPETITION IN SPARSE BINARY PROJECTIONS

## ABSTRACT

Inspired by the advances in biological science, the study of sparse binary projection models has attracted considerable recent research attention. The models project dense input samples into a higher-dimensional space and output sparse binary data representations after Winner-Take-All competition, subject to the constraint that the projection matrix is also sparse and binary. Following the work along this line, we developed a supervised-WTA model when training samples with both input and output representations are available, from which the *optimal* projection matrix can be obtained with a simple, efficient yet effective algorithm. We further extended the model and the algorithm to an unsupervised setting where only the input representation of the samples is available. In a series of empirical evaluation on similarity search tasks, the proposed models reported significantly improved results over the state-of-the-art methods in both search accuracy and running time. The successful results give us strong confidence that the work provides a highly practical tool to real world applications.

## 1 INTRODUCTION

Random projection has emerged as a powerful tool in data analysis applications (Bingham & Mannila, 2001). It is often used to reduce the dimension of a set of data samples in the Euclidean space. It provides a simple and computationally efficient way to reduce the storage complexity of the data by trading a controlled amount of representation error for faster processing speed and smaller model sizes (Johnson & Lindenstrauss, 1984).

Very recently, with strong biological evidence, a sparse binary projection model called the FLY algorithm was designed and attracted people's much attention. Instead of performing dimension reduction, the algorithm increases the dimension of the input samples with a random sparse binary projection matrix. After winner-take-all (WTA) competition in the output space, the samples are converted into a set of sparse binary vectors. In similarity search tasks, it was reported that such sparse binary vectors outperformed the hashed vectors produced by the classical locality sensitive hashing (LSH) method that is based on the random dense projection (Dasgupta et al., 2017).

Following the work along this line, we proposed two models with the explicit treatment of the WTA competition. Instead of residing on the random generation of the projection matrix, one of our models seeks the *optimal* projection matrix under a supervised setting, while the other model operates purely in an unsupervised manner. For each model, we derived an algorithm that is surprisingly simple, yet with highly promising empirical results in both similarity search accuracy and running speed over the state-of-the-art approaches.

A note on notation. Unless specified otherwise, a capital letter, such as $W$, denotes a matrix. A lower-cased letter, with or without a subscript, denotes a vector or a scalar. For example $w_{i\cdot}$ denotes the $i$-th row, $w_{\cdot j}$ denotes the $j$-th column, and $w_{ij}$ denotes the $(i, j)$-th entry of the matrix $W$.

The paper is organized as follows. Section 2 introduces the necessary background. Section 3 presents our models and the algorithms. Section 4 reports the experiments and results, followed by the discussion and conclusion in Section 5.

## 2 BACKGROUND

### 2.1 SPARSE BINARY PROJECTION ALGORITHMS

Different from classical projection methods that commonly map data from a higher-dimensional space to a lower-dimensional space, the FLY algorithm increases the dimension of the data. It was designed by simulating the fruit fly's olfactory circuit, whose function is to associate similar odors with similar tags. Each odor is initially represented as a $50$-dimensional feature vector of firing rates. To associate each odor with a tag involves three steps. Firstly, a divisive normalization step (Olsen et al., 2010) centers the mean of the feature vector. Secondly, the dimension of the feature vector is expanded from 50 to $2,000$ with a sparse binary connection matrix (Caron et al., 2013; Zheng et al., 2018), which has the same number of ones in each row. Thirdly, WTA competition is involved as a result of strong inhibitory feedback coming from an inhibitory neuron. After the competition, all but the highest-firing $5\%$ out of the $2,000$ features are silenced (Stevens, 2015). These remaining $5\%$ features just correspond to the tag assigned to the input odor.

The FLY algorithm can be studied as a special form of the locality sensitive hashing (LSH) method which produces similar hashes for similar input samples. But different from the classical LSH method which reduces the data dimension, the FLY algorithm increases the dimension with a random sparse binary matrix, while ensuring the sparsity and binarization of the data in the output space. Empirically, the FLY algorithm reported improved results over the LSH method (Dasgupta et al., 2017) in similarity search applications.

The success of the FLY algorithm inspired considerable research attention, among which one of particular interest to us is the LIFTING algorithm (Li et al., 2018) that removes the randomness assumption of the projection matrix, which is partially supported by most recent biological discoveries (Zheng et al., 2018). In the work, the projection matrix is obtained through supervised learning. Suppose training samples with both dense input representation $X \in \mathcal{R}^{d \times m}$ and sparse output representation $Y \in \{0, 1\}^{d' \times m}$ are available. The LIFTING algorithm seeks the projection matrix $W$ that minimizes $\|WX - Y\|_F^2 + \beta \|W\|_{\frac{1}{2}}$ in the feasible region of sparse binary matrices. To solve the optimization problem, the Frank-Wolfe algorithm was suggested with quite good performance (Frank & Wolfe, 1956; Jaggi, 2013).

### 2.2 WINNER-TAKE-ALL COMPETITION

Evidences in neuroscience showed that excitation and inhibition are common activities in neurons (Stevens, 2015; Turner et al., 2008). Based on the lateral information, some neurons raise to the excitatory state; while the rest get inhibited and remain silent. The excitation and inhibition result in the competition among neurons. Modeling such neuron competitions is of key importance. Computational models that involve the competition stage are found in wide applications, such as in computational brain models, artificial neural networks and analog circuits design (Arbib, 2003; Maass, 2000). In machine learning, the competition mechanism has motivated the design of computer algorithms for a long time, from the early self-organizing map (Kohonen, 1990) to more recent work in developing novel neural network architectures (Panousis et al., 2019; Lynch et al., 2019).

To model the competition stage, the WTA model is routinely adopted. We are interested with a variant of the WTA model with the following form. For a $d$-dimensional input vector $x$ and a given hash length $k$ ($k \ll d$), the WTA competition computes a function $WTA_k^d : \mathcal{R}^d \to \{0, 1\}^d$ with the output $y = WTA_k^d(x)$ satisfying, for each $1 \le i \le d$,

$$y_i = \begin{cases} 1, & \text{if } x_i \text{ is among the } k\text{-largest entries of } (x_1, \cdots, x_d). \\ 0, & \text{otherwise.} \end{cases} \quad (1)$$

Thus the output entries with value 1 just mark the positions of the $k$-largest values of $x$. For simplicity and without causing ambiguity, we will not differentiate whether the input/output vector of the $WTA$ function is a row vector or a column vector.

## 3 MODEL

### 3.1 SUPERVISED TRAINING

Let a set of samples be in the form of $X \in \mathcal{R}^{d \times n}$ and $Y \in \{0, 1\}^{d' \times n}$ with each $x_{.m}$ $(1 \leq m \leq n)$ being an input sample and $y_{.m}$ being its output representation satisfying $\|y_{.m}\|_1 = k$ for a given $k$. We assume that there exists a projection matrix $W \in \{0, 1\}^{d' \times d}$ with each $\|w_{i.}\|_1 = c$ $(1 \leq i \leq d')$ for a given $c$ [1], such that each $y_{.m} = WTA_k^{d'}(Wx_{.m})$ holds.

From the definition of the WTA function in Eq. (1), a necessary and sufficient condition for a sparse binary matrix $W$ being the projection matrix for the given samples is:

$$w_{i.}x_{.m} \geq w_{j.}x_{.m}, \text{ if } y_{im} = 1 \text{ and } y_{jm} = 0 \tag{2}$$

for all $1 \leq m \leq n$ and $1 \leq i, j \leq d'$.

In a supervised setting, we are interested in seeking such a projection matrix $W$. But unfortunately, solving the problem directly from Eq. (2) is generally hard. A matrix that satisfies all the constraints may not exist for the data due to the noise in observation. Even if it exists, the computational requirement can be non-trivial. A straightforward approach, modeling the problem as a linear integer program, would involve $d' \times d$ variables and $O(nk(d' - k) + d')$ constraints, which is infeasible to solve even for moderately small $n$ and $d'$.

To handle the difficulty, the problem can be relaxed. For each $i, j$ and $m$, define a measure $y_{im}(1 - y_{jm})(w_{i.}x_{.m} - w_{j.}x_{.m})$ to quantify the compliance with the condition in Eq. (2). For $y_{im} = 1$ and $y_{jm} = 0$, the measure has a non-negative value if the condition is met; otherwise, it is negative. Naturally, we sum up the values of the measure over all possible $m, i, j$, and define

$$L_s(W) = \sum_{m=1}^{n} \sum_{i=1}^{d'} \sum_{j=1}^{d'} y_{im}(1 - y_{jm})(w_{i.}x_{.m} - w_{j.}x_{.m}). \tag{3}$$

The value of $L_s(W)$ measures how well a matrix $W$ meets the conditions in Eq. (2). Maximizing $L_s$ with respect to $W$ in the feasible region of sparse binary matrices provides a principled solution to seeking the projection matrix. And we call this the supervised-WTA model.

Considering that

$$
\begin{aligned}
\max L_s(W) &\iff \max \sum_{m=1}^{n} \left[ d' \sum_{i=1}^{d'} y_{im}w_{i.}x_{.m} - k \sum_{j=1}^{d'} w_{j.}x_{.m} \right] \\
&\iff \max \sum_{m=1}^{n} \left[ \sum_{i=1}^{d'} y_{im}w_{i.}x_{.m} - \frac{k}{d'} \sum_{i=1}^{d'} w_{i.}x_{.m} \right] \\
&\iff \max \sum_{m=1}^{n} \left[ \sum_{i=1}^{d'} \left( y_{im} - \frac{k}{d'} \right) w_{i.}x_{.m} \right] \\
&\iff \sum_{i=1}^{d'} \max \left\{ w_{i.} \left[ \sum_{m=1}^{n} x_{.m} \left( y_{im} - \frac{k}{d'} \right) \right] \right\}
\end{aligned}
$$

Therefore, maximizing $L_s(W)$ is equivalent to $d'$ maximization sub-problems. Each sub-problem seeks a row vector $w_{i.}$ $(1 \leq i \leq d')$ by

$$\max w_{i.} \left[ \sum_{m=1}^{n} x_{.m} \left( y_{im} - \frac{k}{d'} \right) \right] \tag{4}$$

subject to:

$$w_{i.} \in \{0, 1\}^{1 \times d}, \text{ and } \|w_{i.}\|_1 = c. \tag{5}$$

---

[1]Following the work of (Dasgupta et al., 2017), $c$ is set to $\lfloor 0.1 \times d \rfloor$ in this paper.

Let

$$\ell_{.i} = \sum_{m=1}^{n} x_{.m} \left( y_{im} - \frac{k}{d'} \right),$$ (6)

and the optimal solution of $w_{i.}$ to Eq. (4) is just given by

$$w_{i.}^* = WTA_c^d \left( \ell_{.i} \right).$$ (7)

## 3.2 UNSUPERVISED TRAINING

The supervised-WTA model utilizes both input and output representations to learn a projection matrix. When only the input representation is available, we can extend the work to an unsupervised-WTA model, by maximizing the objective:

$$L_u \left( W, Y \right) = \sum_{m=1}^{n} \sum_{i=1}^{d'} \sum_{j=1}^{d'} y_{im} \left( 1 - y_{jm} \right) \left( w_{i.} x_{.m} - w_{j.} x_{.m} \right)$$ (8)

satisfying

$$w_{i.} \in \{0, 1\}^{1 \times d}, \|w_{i.}\|_1 = c, y_{.m} \in \{0, 1\}^{d' \times 1}, \text{ and } \|y_{.m}\|_1 = k$$ (9)

for all $1 \leq i \leq d'$ and $1 \leq m \leq n$.

Different from the supervised model, this unsupervised model treats the unknown output representation $Y$ as a variable, and jointly optimizes on both $W$ and $Y$. To maximize $L_u$, an alternating algorithm can be used. Start with a random initialization of $W$ as $W^1$, and solve the model iteratively. In $t$-th ($t = 1, 2, \cdots$) iteration, maximize $L_u \left( W^t, Y \right)$ with respect to $Y$ and get the optimal $Y^t$. Then maximize $L_u \left( W, Y^t \right)$ with respect to $W$ and get the optimal $W^{t+1}$.

The optimal $Y^t$ is given by:

$$y_{.m}^t = WTA_k^{d'} \left( W^t x_{.m} \right)$$ (10)

for all $1 \leq m \leq n$. Similarly to the supervised setting, the optimal $W^{t+1}$ is given by:

$$w_{i.}^{t+1} = WTA_c^d \left( \ell_{.i}^t \right)$$ (11)

for all $1 \leq i \leq d'$, where $\ell_{.i}^t = \sum_{m=1}^{n} x_{.m} \left( y_{im}^t - \frac{k}{d'} \right)$.

Denote by $L_u^t = L_u \left( W^t, Y^t \right)$. Obviously, the sequence $\{L_u^t\}$ monotonically increases for $t = 1, 2, \cdots$. Therefore the alternating optimization process is guaranteed to converge when the objective value $L_u^t$ can't be increased any more.

It is worth mentioning that the unsupervised-WTA model can be studied as a clustering method (Jain et al., 1999). The model puts $m$ data samples into $d'$ clusters and each sample belongs to $k$ clusters. A special case of $k = 1$ corresponds to a hard clustering method. Two samples with the element of one in the same output dimension indicates that they have the same cluster membership.

The unsupervised-WTA model can also be treated as a feature selection method (Guyon & Elisseeff, 2003). This can be seen from the fact that each output dimension is associated with a subset of $c$ features, instead of all $d$ features in the input space. The model is able to choose these $c$ features automatically and encode the information in the projection matrix $W$. A detailed discussion of the clustering and the feature selection viewpoints goes beyond the scope of this paper and is hence omitted.

## 3.3 COMPLEXITY ISSUES

Solving the supervised-WTA model is computationally straightforward and can be implemented with high efficiency. For each projection vector $w_{i.}$, a naïve implementation needs $O \left( dn + d \log c \right)$ operations, among which $O \left( dn \right)$ are for the summation operation in Eq. (6) and $O \left( d \log c \right)$ are for the sorting operations in Eq. (7). Therefore, computing the whole projection matrix needs $O \left( d'dn + d'd \log c \right)$ operations. In fact, by utilizing the sparse structure of the output matrix $Y$, the computational complexity for $W$ can be further reduced to $O \left( kdn + d'd \log c \right)$. As seen in Section 4.3, this is a highly efficient result.

To solve the unsupervised-WTA model, each iteration we need to compute both $Y$ and $W$. Solving one $Y$ needs $O\left(cdn + d'd\log k\right)$ operations by utilizing the sparse structure of $W$, where $O\left(cdn\right)$ are for multiplying $W$ with $X$ and $O\left(d'd\log k\right)$ are for the sorting operations in Eq. (10). Computing one $W$ has the same complexity as in the supervised-WTA model, $O\left(kdn + d'd\log c\right)$. Therefore, the total complexity per iteration is $O\left((k+c)dn + d'd\log(kc)\right)$, which is also an efficient solution as seen in Section 4.3.

For both WTA models, the memory requirement comes mainly from the matrices $X$, $Y$ and $W$, and the storage complexity is $O\left(dn + d'n + d'd\right)$, which can be further reduced to $O\left(dn + kn + d'c\right)$ if sparse matrix representation is adopted.

The training algorithms are parallelizable. Each vector of $W$ and $Y$ can be solved independently with high parallel efficiency. It is also notable that, after simple pre-processing of the training data, all computations only involve simple vector addition and scalar comparison operations.

# 4 EVALUATION

## 4.1 GENERAL SETTINGS

To evaluate the performance of the proposed models, we carried out a series of experiments under the following settings.

**Application:** Similarly to the work of (Dasgupta et al., 2017), we applied the proposed models in similarity search tasks. Similarity search aims to find similar samples to a given query object among potential candidates, according to a certain distance or similarity measure (Baeza-Yates & Ribeiro-Neto, 1999). The complexity of accurately determining similar samples relies heavily on both the number of candidates and their dimension. Computing the distances seems straightforward, but unfortunately could often become prohibitive if the number of candidates is too large or the dimension of the data is too high.

To handle the difficulty brought by the high dimension of the input data, we can either reduce the data dimension while approximately preserving their pairwise distances, or increase the dimension but confining the data in the output space to be sparse and binary, in the hope of significantly improved search speed with the new representation.

**Objective:** Our major objective is to evaluate and compare the similarity search accuracies for different algorithms. Each sample in a given dataset was used, in turn, as the query object, and the rest samples in the same dataset were used as the search candidates. For each query object, we compared its 100 nearest neighbors in the output space with its 100 nearest neighbors in the input space, and recorded the ratio of common neighbors in both spaces. The ratio is averaged over all query objects as the search accuracy of each algorithm. Obviously, a higher similarity search accuracy indicates a better preserving of locality structures from the input space to the output space by the algorithm.

**Datasets:** In the evaluation, three real datasets and five artificially generated datasets were used. The real datasets have the input representation $X$ only; while the artificial datasets have both the input representation $X$ and the ground-truth of the output representation $Y$. The datasets are:

1. GLOVE: 100- to 1000-dimensional *GloVe* word vectors (Pennington et al., 2014) trained on a subset of 330 million tokens from wikimedia database dumps[2] with the $50,000$ most frequent words.
2. MNIST: 784-dimensional images of handwritten digits in gray-scale (LeCun et al., 1998).
3. SIFT: 128-dimensional SIFT descriptors of images used for similarity search (Jegou et al., 2011).
4. ARTIFICIAL: five sets of $1,000$-dimensional dense vectors ($X$) and $2,000$-dimensional sparse binary vectors ($Y$). For each hash length of $k = 2/4/8/16/32$, a set of $2,000$-dimensional sparse binary vectors were randomly generated with the hash length. Then the vectors were projected to $1,000$-dimensional dense vectors through principal component analysis. In this way, the samples' pairwise distances are roughly preserved between the input space and the output space; i.e., $\left\|x_{.m} - x_{.m'}\right\|^2 \approx \left\|y_{.m} - y_{.m'}\right\|^2$ for all pairs of samples in the same set.

---

[2]https://dumps.wikimedia.org/

**Algorithms to compare:** We compared the proposed supervised-WTA (denoted by SUP) model and the unsupervised-WTA (UNSUP) model with the LSH algorithm (Gionis et al., 1999; Charikar, 2002), the FLY algorithm (Dasgupta et al., 2017) and the LIFTING algorithm (Li et al., 2018). The LSH algorithm maps $d$-dimensional inputs to $k$-dimensional dense vectors with a random dense projection matrix. The FLY algorithm uses a random sparse binary matrix to map $d$-dimensional inputs to $d'$-dimensional vectors. The LIFTING algorithm trains a sparse binary projection matrix in a supervised manner for the $d$-dimensional to $d'$-dimensional projection. Both the FLY and the LIFTING algorithms involve a WTA competition stage in the output space to generate sparse binary vectors for each hash length $k$.

Besides, we included the comparison with a number of more recent hashing algorithms, including iterative quantization (ITQ) (Gong et al., 2012), spherical hashing (SPH) (Heo et al., 2015) and isotrophic hashing (ISOH) (Kong & Li, 2012). These algorithms were popularly used in literature to produce sparse binary data embeddings.

**Computing environment:** All the algorithms were implemented in MATLAB platform running on an 8-way server, with which 128 threads were enabled for each algorithm. For the LIFTING algorithm, IBM CPLEX was used as the linear program solver that is needed by the Frank-Wolfe algorithm.

## 4.2 SIMILARITY SEARCH ACCURACY

Table 1: Comparison of similarity search accuracies on various datasets with fixed output dimension ($d' = 2,000$).

| Datasets | $k$ | SUP | UNSUP | LSH | FLY | LIFTING | ITQ | SPH | ISOH |
|---|---|---|---|---|---|---|---|---|---|
| | 2 | **0.1758** | 0.1143 | 0.0174 | 0.0474 | 0.1748 | 0.0103 | 0.0097 | 0.0101 |
| ARTIFICIAL | 4 | **0.6665** | 0.3531 | 0.0243 | 0.0673 | 0.6134 | 0.0175 | 0.0138 | 0.0227 |
| $d = 1,000$ | 8 | 0.3647 | **0.3944** | 0.0259 | 0.0376 | 0.2612 | 0.0360 | 0.0173 | 0.0331 |
| | 16 | **0.5884** | 0.3267 | 0.0278 | 0.0402 | 0.1694 | 0.0367 | 0.0202 | 0.0349 |
| | 32 | **0.3141** | 0.1319 | 0.0324 | 0.0443 | 0.0832 | 0.0382 | 0.0235 | 0.0375 |
| | 2 | 0.1317 | **0.1596** | 0.0217 | 0.0511 | 0.0831 | 0.0221 | 0.0195 | 0.0198 |
| GLOVE | 4 | 0.2310 | **0.3251** | 0.0356 | 0.0964 | 0.1458 | 0.0617 | 0.0311 | 0.0594 |
| $d = 300$ | 8 | 0.3061 | **0.3959** | 0.0655 | 0.1073 | 0.1914 | 0.1209 | 0.0591 | 0.1112 |
| | 16 | 0.4030 | **0.4495** | 0.1138 | 0.1809 | 0.2851 | 0.1939 | 0.1004 | 0.1882 |
| | 32 | **0.4374** | 0.4323 | 0.2039 | 0.2808 | 0.3917 | 0.3208 | 0.1717 | 0.2727 |
| | 2 | **0.2860** | 0.2476 | 0.0369 | 0.1119 | 0.1159 | 0.0288 | 0.0237 | 0.0194 |
| MNIST | 4 | 0.3338 | **0.3829** | 0.0844 | 0.1721 | 0.2003 | 0.0852 | 0.0649 | 0.0773 |
| $d = 784$ | 8 | 0.3885 | **0.4387** | 0.1823 | 0.2717 | 0.3044 | 0.2008 | 0.1443 | 0.1601 |
| | 16 | 0.4698 | **0.4957** | 0.3226 | 0.3953 | 0.4150 | 0.3207 | 0.2559 | 0.3101 |
| | 32 | 0.5108 | **0.5207** | 0.4773 | 0.4773 | 0.5130 | 0.4415 | 0.3616 | 0.4067 |
| | 2 | **0.1706** | 0.1502 | 0.0355 | 0.1066 | 0.1139 | 0.0274 | 0.0272 | 0.0230 |
| SIFT | 4 | 0.2240 | **0.2278** | 0.0760 | 0.1592 | 0.2120 | 0.0550 | 0.0596 | 0.0623 |
| $d = 128$ | 8 | 0.3768 | **0.3912** | 0.1556 | 0.2382 | 0.3059 | 0.0951 | 0.1153 | 0.1240 |
| | 16 | 0.4353 | **0.4461** | 0.2751 | 0.3409 | 0.3529 | 0.1712 | 0.1993 | 0.1905 |
| | 32 | **0.4839** | 0.4751 | 0.4122 | 0.4504 | 0.4295 | 0.3217 | 0.2582 | 0.2832 |

Table 2: Comparison of similarity search accuracies on GLOVE dataset with various input dimensions and fixed output dimension ($d' = 2,000$).

| Dimension | $k$ | SUP | UNSUP | LSH | FLY | LIFTING | ITQ | SPH | ISOH |
|---|---|---|---|---|---|---|---|---|---|
| | 2 | 0.1007 | **0.1210** | 0.0208 | 0.0503 | 0.0982 | 0.0245 | 0.0229 | 0.0230 |
| | 4 | 0.1720 | **0.2274** | 0.0335 | 0.0787 | 0.1449 | 0.0683 | 0.0452 | 0.0633 |
| $d = 100$ | 8 | 0.2365 | **0.2816** | 0.0591 | 0.1059 | 0.1898 | 0.1509 | 0.0705 | 0.1297 |
| | 16 | 0.3113 | **0.3572** | 0.1096 | 0.1698 | 0.2279 | 0.2201 | 0.1232 | 0.1995 |
| | 32 | 0.3779 | **0.3831** | 0.1962 | 0.2581 | 0.2954 | 0.3311 | 0.2398 | 0.2952 |
| | 2 | 0.0808 | **0.1073** | 0.0183 | 0.0387 | 0.0733 | 0.0231 | 0.0223 | 0.1234 |
| | 4 | 0.1432 | **0.2030** | 0.0275 | 0.0624 | 0.1037 | 0.0692 | 0.0395 | 0.0212 |
| $d = 200$ | 8 | 0.2008 | **0.2551** | 0.0459 | 0.0786 | 0.1363 | 0.1197 | 0.0624 | 0.1256 |
| | 16 | 0.2759 | **0.3189** | 0.0816 | 0.1284 | 0.1712 | 0.1804 | 0.1105 | 0.1905 |
| | 32 | 0.3254 | **0.3331** | 0.1442 | 0.1991 | 0.2391 | 0.3025 | 0.2051 | 0.2782 |
| | 2 | 0.0689 | **0.0866** | 0.0148 | 0.0226 | 0.0490 | 0.0197 | 0.0173 | 0.0182 |
| | 4 | 0.1328 | **0.1702** | 0.0195 | 0.0394 | 0.0711 | 0.0522 | 0.0301 | 0.0397 |
| $d = 500$ | 8 | 0.1878 | **0.2252** | 0.0278 | 0.0421 | 0.0892 | 0.0973 | 0.0521 | 0.0885 |
| | 16 | **0.2768** | 0.2696 | 0.0437 | 0.0710 | 0.1188 | 0.1497 | 0.0995 | 0.1305 |
| | 32 | **0.3172** | 0.2727 | 0.0728 | 0.1115 | 0.1806 | 0.2119 | 0.1502 | 0.2117 |
| | 2 | 0.0464 | **0.0508** | 0.0132 | 0.0172 | 0.0293 | 0.0177 | 0.0166 | 0.0179 |
| | 4 | 0.0985 | **0.1131** | 0.0162 | 0.0288 | 0.0396 | 0.0356 | 0.0289 | 0.0322 |
| $d = 1,000$ | 8 | 0.1447 | **0.1615** | 0.0214 | 0.0303 | 0.0490 | 0.0434 | 0.0312 | 0.0365 |
| | 16 | **0.2115** | 0.2071 | 0.0305 | 0.0474 | 0.0662 | 0.0912 | 0.0787 | 0.0883 |
| | 32 | **0.2194** | 0.2049 | 0.0464 | 0.0699 | 0.0874 | 0.1507 | 0.1339 | 0.1303 |

We carried out the experiments on the artificial datasets and the real datasets. From each ARTIFI-CIAL dataset, we randomly chose $10,000$ training samples with both the input ($X$) and the output ($Y$) representations, and chose another $10,000$ testing samples with the input representation only. For the supervised-WTA model, we trained a sparse binary projection matrix $W$, and generated the $2,000$-dimensional sparse binary output vectors for the testing samples with the matrix and after WTA competition. For the LIFTING algorithm, the same training and testing procedures were applied. For all other algorithms, we applied each on the testing samples to get either dense or sparse binary output vectors. Then the output representation is adopted in similarity search and compared against the input representation, as illustrated in Section 4.1.

We repeated the experiments for fifty runs and recorded the average accuracies. In Table 1, each row shows the similarity search accuracies with the output representations generated by the algorithms with the same hash length[3]. Consistent with the results reported in (Dasgupta et al., 2017), the sparse binary projection algorithms reported improved results over the classical LSH method. It is evidently shown that, with the support of the supervised information, both the LIFTING algorithm and the supervised-WTA model reported further improved results. Most prominently, when the hash length $k = 4$, the FLY algorithm has an accuracy of $6.73\%$, while the supervised-WTA model's accuracy reaches $66.7\%$, almost ten times higher. When comparing the two supervised algorithms, the supervised-WTA model outperformed the LIFTING algorithm with all hash lengths.

When comparing the unsupervised algorithms, the unsupervised-WTA model reported improved results, significantly better than the results on the representations generated by the LSH, FLY, ITQ, SPH and ISOH algorithms. Its performance is even better than the supervised-WTA model when the hash length $k = 8$.

On GLOVE/MNIST/SIFT datasets, only the input $X$ is available. As suggested in (Li et al., 2018), we computed $Y^* = \arg_Y \min \frac{1}{2} \left\| X^T X - Y^T Y \right\|_F^2 + \gamma \left\| Y \right\|_{\frac{1}{2}}$ in the feasible region of sparse binary matrices with the Frank-Wolfe algorithm, and used $Y^*$ as the output representation for the supervised models. We carried out the experiments under the same setting as on the ARTIFICIAL datasets. Again the two WTA models reported evidently improved results.

When comparing the two WTA models, the unsupervised-WTA model seems to perform even better than the supervised-WTA model on these real datasets. In the case with known $X$ only, an approximation of $Y$ has to be obtained through matrix factorization. The quality of this approximated $Y$ becomes critical and sometimes the vulnerability for the supervised algorithms. We believe this is one major reason why supervised-WTA no longer excels.

In addition to the experiment on similarity search accuracies, we further investigated the influence of different input/output dimensions on the performance of the proposed models. In this experiment, we fixed the output dimension to $d' = 2,000$ while varying the input dimension from $100$ to $1,000$ on *GloVe* word vectors. We recorded the similarity search accuracies by all the algorithms. From the results in Table 2, we can see that the WTA models reported consistently improved results under all settings.

## 4.3 RUNNING SPEED

As a practical concern, we measured the training time of our proposed WTA models, and compared it with the LIFTING algorithm. In the experiment, we used the ARTIFICIAL datasets with $1,000$-dimensional inputs and $2,000$-dimensional outputs, and the number of training samples varied from $1,000$ to $50,000$.

We recorded the training time by each algorithm to compute the sparse binary projection matrix $W$. On all training sets, our proposed models reported significantly faster speed over the LIFTING algorithm. With $n = 1,000$ samples (ref. Fig. 1(a)), the supervised-WTA model took less than $0.2$ seconds to get the optimal solution, hundreds of times faster than the LIFTING algorithm which used around $50$ seconds.

---

[3]Following the work of (Dasgupta et al., 2017; Li et al., 2018), the hash length is defined as the number of ones in each output vector for the FLY, LIFTING and WTA algorithms. For other algorithms, it is defined as the output dimensionality.

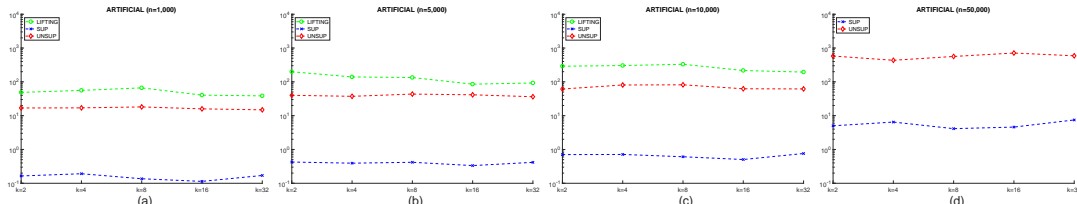

Figure 1: **Comparison of running time on ARTIFICIAL dataset with different number of training samples** ($n = 1,000/5,000/10,000/50,000$). The horizontal axis is the hash length ($k = 2/4/8/16/32$). The vertical axis is the running time (seconds) in log-scale. When $n = 50,000$, the LIFTING algorithm didn't finish within 12 hours and its result is not reported.

The unsupervised-WTA model needs to solve multiple $W^t$ and $Y^t$ iteratively. It spent 10 to 20 seconds, which was slower than the supervised-WTA model but several times faster than the LIFTING algorithm. With $n = 50,000$ training samples (ref. Fig. 1(d)), the supervised-WTA model took less than 10 seconds, and the unsupervised-WTA model took around 400 seconds to get the solutions. While for the LIFTING algorithm, we didn't finish the execution in our platform within 12 hours. All these real results were consistent with the complexity analysis given in Section 3.3, and justified the running efficiency of our proposed WTA models.

## 5 CONCLUSION

With strong support from biological science, the study of sparse binary projection models has attracted much research attention recently. Through converting lower-dimensional dense data to higher-dimensional sparse binary vectors, the models have achieved excellent empirical results and provided a useful tool in practical applications.

Biologically, the winner-take-all competition is an important stage for pattern recognition activities that happen in the brain. Our work started from the treatment of the WTA competition, and proposed two models to seek the desired projection matrix for sparse binary projections. The supervised-WTA model utilizes both input and output representations of the samples, and trains the projection matrix as a supervised learning problem. The unsupervised-WTA model requires the input representation only, which equips the model with wider application scenarios. For the models, we have developed simple, efficient and effective algorithms. In the evaluation, our models significantly outperformed the state-of-the-art methods, in both search accuracy and running time.

Our work potentially triggers a number of topics. Firstly, the algorithms for both models are simple, only involving vector addition and scalar comparison operations. The algorithms are highly parallelizable. Such characteristics make the computing procedures suitable to be implemented with customized hardware such as in the FPGA platform (Omondi & Rajapakse, 2006), which will provide a high-throughput and economical solution for large scale data analysis applications.

Secondly, the unsupervised-WTA model has a natural relationship with the clustering and the feature selection models. It provides a unified framework that combines these two techniques. We believe that investigating new clustering and feature selection applications following this viewpoint is possible. More importantly, we believe that clustering and feature selection may be a potential bridge that helps to make clear why modeling the WTA competition could lead to algorithms that preserve the locality structures of the data well, as reported in this paper.

The third potential is in designing new artificial neural network architectures. The approach adopted in this paper can be used as an activation function of the neurons in an artificial neural network. We highly anticipate future research work along this line (Pehlevan et al., 2018; Lynch et al., 2019).

ACKNOWLEDGMENTS

Suppressed for blind review.

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
