# OpenReview forum: "Modeling Winner-Take-All Competition in Sparse Binary Projections"
_ICLR.cc/2020/Conference — Reject_

### Official Review · AnonReviewer1 · 2019-10-23
**Official Blind Review #1**

**Rating:** 3

**Review:**

The authors developed a supervised-WTA model when training samples with both input and output representations are available, from which the optimal projection matrix can be obtained. The authors further extended the model and the algorithm to an unsupervised setting where only the input representation of the samples is available.

Overall I think this paper is clearly presented with some good experiment results.

I have the following concerns:

1)As for the supervised setting, may I know in what kind of real-world applications we can find the OPTIMAL binary and sparse output representation?  Why can we assume they are optimal?

2)Is this algorithm can really work on a large scale problem? The mnist level dataset is far from giving strong evidence in claiming this point. A larger dataset like ImageNet is preferred.

**Experience Assessment:**

I have published in this field for several years.

**Review Assessment: Checking Correctness Of Derivations And Theory:**

I assessed the sensibility of the derivations and theory.

**Review Assessment: Checking Correctness Of Experiments:**

I assessed the sensibility of the experiments.

**Review Assessment: Thoroughness In Paper Reading:**

I read the paper at least twice and used my best judgement in assessing the paper.

---

> ### Author Response · Authors · 2019-11-15
> **Response to Review #1**
>
>
> 1. About optimal outputs: The supervised learning studied in the paper is different from a typical supervised setting (as commented in Review #3). Instead of real samples, the outputs are from a special projection obtained through matrix factorization.
> Thanks for the suggestion. Real training samples also exist in practice. For example, in multi-label classification (MLC), each output is a binary vector. These ground-truth vectors may be regarded as the optimal output representation. The performance of our proposed model in such MLC applications (http://mulan.sourceforge.net/datasets-mlc.html) may deserve our high attention.
>
> 2. About scalability: Supervised-WTA is a special case of unsupervised-WTA, and we only consider the unsupervised setting here. As shown in Section 3.3, the complexity in each iteration is O((k+c)*d*n + d'*d*log(kc)). The complexity grows linearly with the number of samples n, which should be a reasonable requirement in big data applications.
>
> We tested the algorithm on 10K/50K/100K/500K/1M ImageNet images (http://image-net.org/api/text/imagenet.sbow.obtain_synset_wordlist). When the hash length k=32, the training converged in ~3 minutes for n=10K, ~8 minutes for n=100K and ~100 minutes for n=1M images, which fit the complexity analysis quite well. The following two tables show the training time and testing accuracy.
>
>        n=10K/50K/100K/500K/1M
> k=2:   35.1/126.3/343.2/626.8/893.4
> k=4:   50.6/188.7/321.0/779.2/1030.5
> k=8:   79.3/286.3/376.4/920.0/1912.2
> k=16:  128.2/221.7/280.3/1428.0/2816.1
> k=32:  177.8/277.9/458.1/2674.4/6047.3
> Unsupervised-WTA training time (in seconds) on ImageNet with different number of training samples. A maximum of 128 CPU threads were allowed (d'=2000).
>
>             k=2/4/8/16/32
> WTA(n=10K): 0.1389/0.1899/0.2129/0.2329/0.2407
> WTA(n=50K): 0.1303/0.1872/0.2170/0.2380/0.2446
> WTA(n=100K):0.1234/0.1851/0.2157/0.2383/0.2434
> WTA(n=500K):0.1287/0.1845/0.2183/0.2395/0.2470
> WTA(n=1M):  0.1280/0.1863/0.2177/0.2391/0.2480
> LSH:        0.0251/0.0502/0.0824/0.1522/0.2282
> FJL:        0.0224/0.0578/0.0854/0.1527/0.2337
> FLY:        0.0668/0.1058/0.1519/0.2122/0.2430
> Comparison of unsupervised-WTA with LSH/FJL/FLY based projections on similarity search accuracies of 100 nearest neighbors from 10000 testing samples on ImageNet (d'=2000).

---

### Official Review · AnonReviewer2 · 2019-10-23
**Official Blind Review #2**

**Rating:** 8

**Review:**

This paper proposed a Winner-Take-All model that learns a sparse binary representation for dense vectors. The method learns a binary lifting matrix to project low-dimension vectors into higher dimension, and apply WTA to get a final binary embedding with fixed number of 1s. I find this paper interesting and good extension of exiting lifting method. Please see my detailed comments:

1. The assumption of the existence of W seems strong. I understand in reality it may not hold but shouldn't affect the representation learning and downstream applications of the binary vector, but is there any work done to estimate whether it holds in general or only for d' in some range?

2. In experiments, both the supervised and unsupervised methods are compared. For the supervised version and for search accuracy evaluation, what is the ground truth? E.g. what's the label for Glove data?

3. Is the term in equation (3) first proposed in this paper or already used before?

4. How to choose the value of d' in practice? How to balance the speed and representation quality?

5. Is there an information theoretical way to evaluate whether y has encoded near complete information from the previous vector? Or the high-dimension binary vector only works in practice without deeper theories?

**Experience Assessment:**

I have read many papers in this area.

**Review Assessment: Checking Correctness Of Derivations And Theory:**

I assessed the sensibility of the derivations and theory.

**Review Assessment: Checking Correctness Of Experiments:**

I carefully checked the experiments.

**Review Assessment: Thoroughness In Paper Reading:**

I read the paper at least twice and used my best judgement in assessing the paper.

---

> ### Author Response · Authors · 2019-11-15
> **Response to Review #2**
>
>
> 1. Existence of W: Our assumption of the projection matrix W is mostly motivated by biological evidences. Mathematically for arbitrarily given inputs X and outputs Y, the W may not exist. To address this difficulty, we designed a relaxed model in equation (3). Another potential approach is to borrow the idea of 'soft-margin' in the support vector machines algorithm, which only penalizes the violation samples. Comparatively, our model in equation (3) rewards the correct projections and penalizes the incorrect projections at the same time.
>
> 2. Ground truth of Glove labels: We don't have the ground truth of Glove labels. Instead, we used the Y* (ref. paragraph 4 of page 7) as a kind of pseudo-labels, which were obtained through heavy matrix factorization and fine tuning to ensure X'X and Y'Y are close.
>
> 3. About equation (3): As far as we know, equation (3) is new and firstly proposed in this paper.
>
> 4. On choosing d': Biological evidences from the fruit fly suggest an example of d'=40*d. Our experience about this parameter is mostly empirical. The similarity search accuracy seems to be improved when increasing d', but not often significantly. In practice, we would recommend to set d' from 2*d to 20*d as a balance of the speed and representation quality.

---

### Official Review · AnonReviewer3 · 2019-10-25
**Official Blind Review #3**

**Rating:** 6

**Review:**

The paper takes the binary projection framework used for LSH-type applications (based on Dasgupta et al. 2017) and given training data from that framework, shows an efficient closed-form solution update for the "projection" matrix, and also derives an alternating minimization algorithm for when training data are not available (note that these are not training data in a typical supervised learning sense, i.e., they are not data features, but rather the output of a projection of the very specific type discussed in the paper).

The idea is interesting, and the recent literature suggest that the overall idea works well. This new paper seems to work much better than recent literature (and a lot faster than some), which is very encouraging. The derivation of the closed-form updates is clever and seems to be correct.  Overall, I am quite favorable for this paper.

My chief critical comment is that given the complexity of the model, is is feasible to compare to PCA-based projections as well. That is, usually PCA is not compared with these methods since a random projection method is faster. But this paper proposes data-dependent projection matrices, which is exactly what PCA does.  This paper has been implicitly assuming d << n and trying to avoid the O(n^2) cost of nearest neighbor search. Given that d << n, the complexity of the algorithm in the paper is, roughly (avoiding some logs), O((#iter) d^2 n ), since the constant "c" depends on d, and presumably d' also scales linearly with d. The (#iter) refers to the number of iterations of the alternating minimization. The actual constants are small, but it is still quadratic in d.  PCA takes O(d^2n + d^3) operations when d<n, so therefore PCA is of roughly the same complexity, and thus I'd expect you to compare to it in the numerical comparisons.

With a valid response to the issue of PCA, I would probably "strongly accept" this paper.

(Another note on comparisons: it seems Johnson-Lindenstrauss projections would be applicable, so I'd suggest running a Fast Johnson-Lindenstrauss (Ailon and Chazelle) to compare with also).

Minor notes:
- In section 3.3, how do you get complexities like d log(c) for finding the top c entries? Please cite an algorithm textbook. I might be rusty on this, but the algorithms I can think of to do this are O(  min( d log(d), d c )) operations. Same for the  d' log(k).

- A good number of sentences were grammatically incorrect or sounded awkward, e.g., paragraph 2, "people's much attention"; second sentence in section 2.2, the phrase "; while the rest get inhibited and remain silent" is not a complete phrase by itself (maybe change the semicolon to a comma). Overall, another round of proof reading would help.

**Experience Assessment:**

I have read many papers in this area.

**Review Assessment: Checking Correctness Of Derivations And Theory:**

I carefully checked the derivations and theory.

**Review Assessment: Checking Correctness Of Experiments:**

I assessed the sensibility of the experiments.

**Review Assessment: Thoroughness In Paper Reading:**

I read the paper at least twice and used my best judgement in assessing the paper.

---

> ### Author Response · Authors · 2019-11-15
> **Response to Review #3:**
>
>
> 1. Finding top c entries from d elements: The 'Heapsort' algorithm can be applied with a complexity of d log(c) (Knuth 1997).
>
> 2. Comparison with Fast Johnson-Lindenstrauss Projection (Ailon and Chazelle 2009, Bourgain et al. 2015): Thanks for suggesting the FJL model to us. In our experiments, FJL based projection reported accuracy highly comparable to LSH (ref. results below), yet with much faster retrieval speed than LSH.
>
> 3. Comparison with PCA:
> -On speed:
> Yes, PCA and unsupervised-WTA have comparable training complexity, both analytically and numerically, in computing the projection matrix.
> In retrieval, sparse binary projections (FLY, WTA) are magnitude faster than dense projections (LSH, PCA), as reported in (Dasgupta et al. 2017), as well as faster than FJL projections.
> -On accuracy:
> Similarity search accuracies with PCA based projections are reported below. The results of LSH/FLY/unsupervised-WTA are also copied here from the paper for easy comparison.
> When the hash length k is small (k=2/4/8), our WTA models have significantly better retrieval accuracies than PCA. When k becomes large enough, e.g. k=32, PCA excels on two out of three datasets.
>
> GLOVE(k=2/4/8/16/32)
> LSH:0.0217/0.0356/0.0655/0.1138/0.2039
> FJL:0.0198/0.0328/0.0618/0.1081/0.2139
> FLY:0.0511/0.0964/0.1073/0.1809/0.2808
> WTA:0.1596/0.3251/0.3959/0.4495/0.4323
> PCA:0.0536/0.1202/0.2189/0.3134/0.4136
> Comparison of similarity search accuracies on GLOVE dataset.
>
> MNIST(k=2/4/8/16/32)
> LSH:0.0369/0.0944/0.1823/0.3226/0.4773
> FJL:0.0422/0.1029/0.2004/0.3409/0.4846
> FLY:0.1119/0.1721/0.2717/0.3953/0.5162
> WTA:0.2476/0.3829/0.4387/0.4957/0.5207
> PCA:0.0790/0.2447/0.4040/0.5035/0.6637
> Comparison of similarity search accuracies on MNIST dataset.
>
> SIFT(k=2/4/8/16/32)
> LSH:0.0355/0.0760/0.1556/0.2751/0.4122
> FJL:0.0344/0.0707/0.1692/0.2698/0.4290
> FLY:0.1066/0.1592/0.2382/0.3409/0.4504
> WTA:0.1502/0.2278/0.3912/0.4461/0.4751
> PCA:0.0582/0.1622/0.3139/0.5062/0.6367
> Comparison of similarity search accuracies on SIFT dataset.

---

### Author Response · Authors · 2019-11-15
**Response to All Reviews**

We really appreciate all reviewers' beneficial comments, which indeed help us improve the work. Regarding to the concerns raised in the comments, we wish to take this opportunity to make further clarifications. These clarifications will also be included in the revised paper.

---

### Decision · Program_Chairs · 2019-12-19

**Decision:**

Reject

**Comment:**

This paper proposes a WTA models for binary projection.  While there are notable partial contributions, there is disagreement among the reviewers.   I am most persuaded by the concern expressed that the experiments are not done on datasets that are large enough to be state-of-the-art compared to other random projection investigations.